# Micrometeorological Analysis and Glacier Ablation Simulation in East Kunlun

Weisheng Wang [1], Meiping Sun [1,2,*], Yanjun Che [3], Xiaojun Yao [1], Mingjun Zhang [1] and Shuting Niu [1]

1    College of Geography and Environmental Science, Northwest Normal University, Lanzhou 730070, China; 2021212827@nwnu.edu.cn (W.W.); xj_yao@nwnu.edu.cn (X.Y.); mjzhang@nwnu.edu.cn (M.Z.); 2021212830@nwnu.edu.cn (S.N.)
2    Key Laboratory of Resource Environment and Sustainable Development of Oasis, Lanzhou 730070, China
3    Department of Geographical Science, Yichun University, Yichun 336000, China; cheyanjun@jxycu.edu.cn
*    Correspondence: sunmeiping1982@nwnu.edu.cn

**Abstract:** Worldwide, there are great challenges for meteorological monitoring and glacier ablation monitoring in high-altitude mountain areas. It is often difficult to capture fine-scale climate and glacial changes in high-altitude mountainous areas due to the harsh natural environment and the extreme lack of observational sites. Based on high-altitude meteorological stations erected on the eastern shore of Aqikkule Lake (AQK) and at the terminus of Shenshechuan Glacier (SSG), as well as on mass balance data from SSG, the characteristics and correlation of temperature, solar radiation, relative humidity, precipitation, wind speed and direction of the two regions, and the mass balance in the ablation area of SSG from 30 May 2022 to 18 May 2023 were analyzed, and the average melting depth of SSG was simulated. The results indicate the following: (1) The average annual temperature of AQK and the terminus of SSG is −3.7 °C and −7.7 °C, respectively, and the vertical lapse rate of temperature in the summer half of the year is greater than that in the winter half of the year. Precipitation timing has a great influence on daily temperature differences. (2) Precipitation in both places is concentrated in summer; the glaciers in this area are of the summer recharge type, and precipitation has a significant reducing effect on the solar incident radiation and increases the relative humidity in this region. (3) AQK and SSG both have local circulation development, in the area of AQK all year round due to the lake effect, while the terminus of SSG only has the development of valley winds in the summer, being controlled in the winter by the westerly wind belt. (4) The average mass balance value of the ablation area of SSG was −1786 mm as measured by the range poles method. The average annual ablation depth of SSG simulated by using the empirical formula was 587–597 mm, which is not large compared with other glacier areas in the Tibetan Plateau, and it has the characteristics of typical continental-type glaciers.

**Keywords:** microclimate; glacier; Aqikkule Lake; Shenshechuan Glacier; east Kunlun

## 1. Introduction

Climate change in high-altitude mountain areas is one of the most difficult research objects to grasp, which is not only related to the unique natural conditions of high-altitude mountain areas, but also subject to the scarcity of observation sites [1,2]. At present, most studies use remote sensing data or atmospheric reanalysis data to study climate change in high-altitude mountain areas, but due to the influence of external factors, these data often have large deviations [3,4]. Internationally, the construction of a comprehensive and continuous meteorological monitoring network and micrometeorological observation in high-altitude areas has been encouraged and supported [2], which is of great positive significance for the correct study of the impact of climate change on mountain environments. The cryosphere, dominated by snow and ice, is the most important part of the high-altitude mountain areas, and it is also the sphere that is most affected by climate change. For climate monitoring in high-altitude mountain areas, it is necessary to observe the cryosphere change

to a large extent [5]. In the context of global warming, glaciers are generally retreating, and changes in glacier water resources will have a profound impact on the ecological environment and social economy in cold and arid regions [6–11]. On a global scale, glacier change has become a comprehensive research topic related to the environment and social economy [12–18]. Glaciers are an indicator of climate change, and climate change in glacier areas affects the ablation and accumulation of glaciers [19]. Water (precipitation) and heat (air temperature) conditions are the two main factors for glacier development. Precipitation is the material basis for glacier development, but glaciers can only be formed under certain low-temperature conditions. Therefore, it is of great significance to study the water and heat conditions in glacier areas for glacier mass balance [20]. The Tibetan Plateau (TP) is the main body of Western China, with high terrain and a cold climate. It is the region with the most developed mountain glaciers in the middle and low latitudes in the world, and it is known as the world's "third pole". Its glaciers are crucial to the water supply in the region, known as the Asian water tower [8,21]. In the context of climate warming, the global mountain glaciers, including the Tibetan Plateau, are generally shrinking and thinning, and their mass continues to decrease [22,23]. With climate change and glacier retreat, it is expected that glacier runoff will continue to rise for a certain period of time, and after reaching the maximum the runoff will steadily decline [6,7,10–12], bringing major opportunities and challenges to arid and semi-arid areas that rely mainly on snow and ice meltwater [24]. Under this background, it is of great scientific and practical significance to study the mass balance of glaciers and observe the meteorological elements in glacier areas. The objectives of this study were to elucidate the meteorological characteristics of the high-altitude mountainous areas of east Kunlun using field-monitored meteorological data, and to utilize these data for glacier ablation simulations.

The glacier water resources on the north slope of the Kunlun Mountains nourish the oases on the southern edge of Tarim Basin, and they play an important role in the economic development and ecological security of the southern Xinjiang region of China. In west Kunlun, unlike other regions, the change in glacier mass shows a weak increase or a stable trend, thus attracting wide attention [25–31]. On the other hand, the observational study of east Kunlun is extremely lacking. Ulugh Muztagh Peak is the highest peak in east Kunlun and the second largest glaciation area in the Kunlun Mountains, with extensive development of modern glaciers [32]. Compared with other glacial areas on the TP, the east Kunlun region is less affected by the Indian monsoon and prevailing west wind, and it is mainly affected by continental climate conditions [26], making the meteorological conditions here more complicated. The field observations and meteorological observations of glaciers in this area are very weak, and only some data estimates can be obtained based on model simulations. However, there may be large deviations in these simulation data. According to relevant studies, in many available grid-based climate datasets, precipitation at high altitudes has been seriously underestimated [3,4].

Based on the above background, and relying on the third comprehensive scientific expedition project in Xinjiang, we installed a set of fully automatic weather stations (CR800, USA) in May 2022 at the terminus of Shenshechuan Glacier (SSG) on the north slope of Ulugh Muztagh Peak in east Kunlun, as well as on the east bank of Aqikkule Lake (AQK). In terms of glacier ablation monitoring, we placed six range poles (A–F) using steam boreholes at the terminus of SSG.

## 2. Micrometeorology and Glacier Melt Observation

### 2.1. Micrometeorological Observation

Ulugh Muztagh Peak (87°23′ E, 36°24′ 50′′ N) is located at the west end of east Kunlun, with an altitude of 6973 m. It is located at the junction of the southwest corner of Altun Mountain Nature Reserve in Ruoqiang County (southern Xinjiang) and Tibet. It is the highest peak in the east Kunlun Mountains and the second largest glaciation area in the Kunlun Mountains [33]. Modern glaciers in the Muztagh Peak region are extremely developed, and the types of glaciers are relatively complete, with 213 glaciers and a total

area of 653.3 km$^2$ [34]. In summer, under the influence of low thermal pressure over the TP, warm air rises under the control of westerly wind circulation and mountainous terrain, generating precipitation and supplying glaciers [4]. AQK is located in the southeast of Ruoqiang County, at the northern foot of the Kunlun Mountains and within the Altun Mountain Nature Reserve. It is a saltwater lake with an area of 350 km$^2$ [35]. The lake is 4240 m above sea level, with an average depth of 8 m and a maximum depth of 25 m, with injections of AQK, etc. There is a significant lake effect in this area, which greatly affects the weather conditions in the surrounding area (Figure 1a). In the east Kunlun Mountains region, in addition to the influence of the westerly and monsoon winds, the seasonal variations in the thermal properties of the plateau itself make the meteorological characteristics of the region even more unique [36].

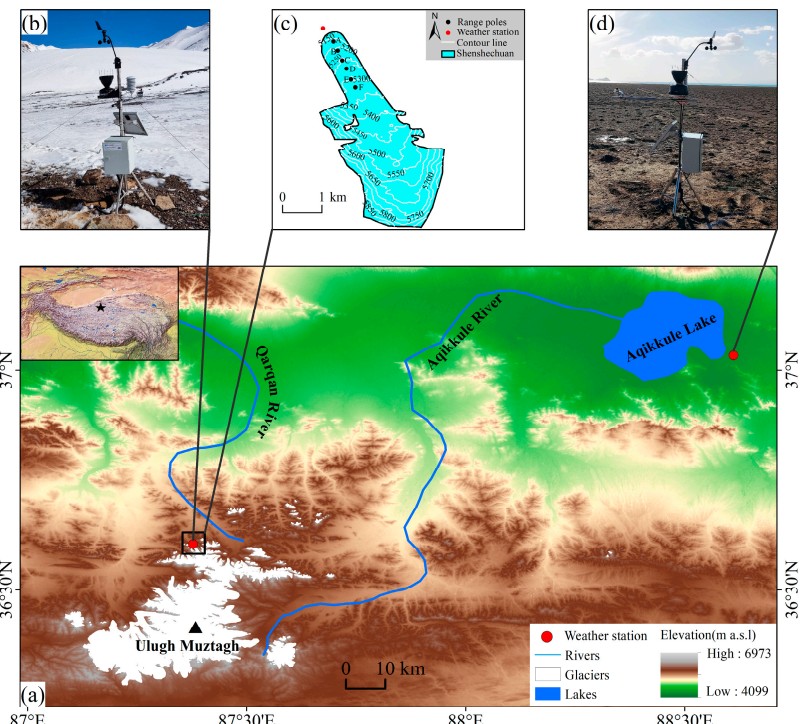

**Figure 1.** Observation of micrometeorology and glacier ablation in east Kunlun: (**a**) The region of the Ulugh Muztagh Peak and AQK. (**b**) The meteorological station at the terminus of SSG. (**c**) Observations of glaciers and glacier ablation for SSG. (**d**) The meteorological station at AQK.

On 28 May 2022, the AQK meteorological station (37.033° N, 88.611° E) was deployed on the east bank of AQK (altitude 4243 m), and the SSG terminal meteorological station (36.605° N, 87.378° E) was deployed on the terminus of SSG (altitude 5070 m) two days later, on May 30. The meteorological elements monitored by the meteorological stations at AQK (Figure 1d) and at the terminus of SSG (Figure 1b) included temperature, solar radiation, relative humidity, precipitation, and wind speed and direction. From 30 May 2022 onwards, meteorological data were monitored at intervals of 10 min, 1 h, and 1 day.

## 2.2. Glacier Ablation Observation

SSG (87°23′15.038″ E, 36°34′51.939″ N) originated from the north slope of Ulugh Muztagh Peak, east Kunlun. The terminus of the glacier is 5100 m above sea level, it is about 5 km long, the total area is 6.05 km$^2$, and the ice reserves are 0.52 km$^3$. The glacier's surface is gentle, with no surface moraine cover [37]. During the observation period from May 2022 to May 2023, the total precipitation at the terminus of the glacier was 415 mm; the maximum precipitation in one day was 22.8 mm. Local circulation developed at the terminus of the glacier. The method used to monitor the glacier mass balance was range

poles. On 30 May 2022, six range poles were arranged by steam-drilling holes in the melting area of the glacier at an altitude of 5200 m to 5300 m. These range poles were named A, B, C, D, E, and F, from low altitude to high altitude (Figure 1c). The data measured by the range poles were collected on 18 May 2023.

## 3. Glacier Ablation Simulation Method

In high and cold regions, glacier mass balance observations are very limited, and glacier ablation modeling has been an important tool to study the changes in glacier mass balance. Glacier ablation models are mainly classified into two categories: the energy balance model, based on physical processes; and the degree-day factors model, based on statistics. The simple degree-day factors model was chosen because other parameters were not available due to the environment of the observation site. The energy balance model, which is based on the interaction between the atmosphere and ice/snow, has a good physical basis and simulation performance in obtaining the heat consumption term used for ice/snow ablation through the quantitative relationship between the various components of energy, such as net radiation, sensible and latent heat fluxes, precipitation, heat flux, and heat flux from the ground. However, due to the large number of input parameters and the spatial distribution of the driving data, it is difficult to achieve the required accuracy of the model; therefore, the generalization of the model to the watershed scale is very limited. So, empirical formulae for glacier ablation at the watershed scale have been more frequently used. There are many empirical formulae for estimating the depth of glacier ablation, most of which are based on the basic principle that the depth of glacier ablation decreases with the increase in altitude. The accumulation of the glacier equilibrium line is equal to the melting amount, and the melting depth at the height of the equilibrium line represents the average melting depth of the glacier.

### 3.1. Climate Coefficient Method

This method is based on the statistical relationship between observed ablation data and surface air temperature on several glaciers in Western China that have been studied. This formula can reflect regional climate differences and can be used to calculate the depth of glacier ablation in regions without data [38].

$$A = 0.382b^2(T + 4.0)^{2.7} \tag{1}$$

where $A$ is the average daily glacier ablation (mm/d), $b$ is the relative value of the glacier radiative balance (%), and $T$ is the average daily air temperature (°C) at the height of the median glacier area during the ablation period.

### 3.2. Glacier Zero-Balance Line Method

The glacier zero-balance line method is a method to estimate the average melting of a glacier based on the basic principle that the accumulation amount and the melting amount on the zero-balance line of the glacier are equal [39]. This formula is widely used in the world, known as the "global formula", and has good application in mountain glaciers and polar glaciers, or marine glaciers and continental glaciers, and the verification effect is good.

$$h = 1.33(t_s + 9.66)^{2.85} \tag{2}$$

where $h$ is the depth of glacial ablation and $t_s$ is the average summer (June–August) temperature (°C) at the balance line.

### 3.3. Degree-Day Factors Method

The degree-day factors method is established on the physical basis of the close relationship between glacier ice/snow ablation and air temperature, especially the positive accumulated temperature on the surface of the ice and snow. Although this model is a simplified description of the complex process of the energy balance of ice and snow abla-

tion, it can give ideal output results similar to the energy balance model on the watershed scale [40].

In many studies, degree-day models are generally of the following form:

$$M = DDF \cdot PDD \tag{3}$$

where $M$ is the water equivalent (mm w.e.) of glacier or snow melt in a given time period, $DDF$ is the degree-day factor for ice or snow (mm d$^{-1}$ °C$^{-1}$), and $PDD$ is the positive cumulative temperature in a given time period.

The positive cumulative temperature is calculated as follows:

$$PDD = \sum_{t=1}^{n} H_t \cdot T_t \tag{4}$$

where $T_t$ is the daily average temperature (°C) of a day ($t$), and $H_t$ is a logical variable; when $T_t \geq 0$ °C, $H_t = 1$; when $T_t < 0$ °C, $H_t = 0$.

Due to the small number of glaciers with long-term observation in the TP and its surrounding areas, it is impossible to calculate the degree-day factor value of each glacier through the observation data of the glacier area. In this study, degree-day factors at the median height of SSG were calculated based on the degree-day factors conversion formula, which is an empirical formula obtained by collecting glacier observation data from different periods in the past decades (Table 1). The degree-day factors of each glacier can be calculated by this formula, which can provide parameter support for regional mass balance simulation and runoff estimation.

**Table 1.** Degree-day factors conversion formula.

| Parameters | Range of Values | Average Value | Formula | r |
|---|---|---|---|---|
| $DDF_{ice}$ | 2.6–16.9 | 7.64 | $DDF_{ice} = 15.763 - 0.277Lat + 0.047Lon - 1.72 \times 10^{-3}H - 0.62T + 6.99 \times 10^{-3}P$ | 0.62 |
| $DDF_{snow}$ | 1.5–9.2 | 4.63 | $DDF_{snow} = 64.533 - 0.837Lat - 0.238Lon - 2.85 \times 10^{-3}H - 1.092T + 2.822 \times 10^{-3}P$ | 0.92 |

Notes: $DDF_{ice}$ and $DDF_{snow}$ denote the degree-day factors (mm d$^{-1}$ °C$^{-1}$) for glacier ice and snow, respectively, while *Lat*, *Lon*, *H*, *T*, and *P* denote glacial latitude, longitude, the terminus of glacial elevation (m), mean annual temperature (°C), and precipitation (mm), respectively.

## 4. Results

### 4.1. Hydrothermal Conditions

4.1.1. Temperature

The changes in temperature in the glacier area will affect the change in the mass balance of the glacier, especially the change in the ablation of the glacier in summer. From Figure 2a, it can be seen that the air temperature at AQK and the terminus of SSG has obvious daily variation, which is in the form of single peak and a single valley, and the trend of change is consistent. The lowest daily temperature at AQK and the terminus of SSG both appeared at 7:00 a.m., with temperatures of −9.4 °C and −10.8 °C, respectively, at which time the minimum daily difference between the two places was 1.4 °C. From 7:00 a.m. onwards, the temperature of the two places gradually increased, and the temperature at AQK and the terminus of SSG reached its highest values at 17:00 p.m. and 15:00 p.m., with temperatures of 2.8 °C and −4.0 °C, respectively, and the temperature difference between the two highest daily temperatures of the two places was the maximum of 6.8 °C, which was 4.9 times the minimum difference in temperature. The average daily maximum temperature at the terminus of SSG is earlier than that at AQK, which is related to differences in elevation and the underlying surface properties between the two places.

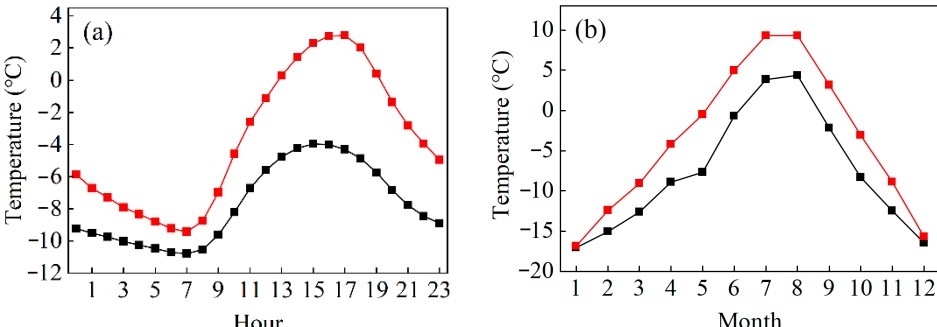

**Figure 2.** Daily (**a**) and monthly (**b**) changes in mean air temperature at AQK and the terminus of SSG; the red line shows the weather station at AQK, while the black line shows the weather station at the terminus of SSG.

On an annual scale (Figure 2b), the annual average temperature at AQK and the terminus of SSG was −3.7 °C and −7.7 °C, respectively. The average monthly minimum temperatures occurred in January, with temperatures of −16.9 °C and −17.1 °C, respectively. The average monthly maximum temperatures occurred in August, with temperatures of 9.3 °C and 4.4 °C, respectively. The average monthly temperatures in the two places decreased month by month after reaching their highest values in August, reached their lowest values in January, and then began to rise. The temperature changes showed obvious seasonal variation. The average temperature at the terminus of SSG was higher than 0 °C from July to August, and the temperature increased the fastest in May to June (7 °C/month), before dropping the fastest in August and September (−6.5 °C/month). The temperature increase rate was slightly greater than the temperature decrease rate. AQK warmed the most in May and June (5.5 °C/month), and the rate of warming was less than that at the terminus of SSG. The difference was that AQK cooled the most in November and December, slightly greater than the terminus of SSG.

The temperature variation of AQK is essentially the same as that at the terminus of SSG, except that due to the altitude, the average daily temperature at the terminus of SSG is lower than that at AQK. In the whole year, the maximum daily average temperatures at AQK and the terminus of SSG occurred on 6 July, which were 14.6 °C and 10.9 °C, respectively, and the minimum daily average temperatures occurred on 17 January, which were −24 °C and −25 °C, respectively. The highest hourly temperatures at AQK and the terminus of SSG occurred at 16:00 and 15:00 on 6 July, when the temperatures were 23.49 °C and 17.81 °C, respectively, and the lowest hourly temperatures occurred at 08:00 on 19 January and 09:00 on 17 January, when the temperatures were −33.62 °C and −28.92 °C, respectively. At this time, AQK was colder than the terminus of SSG, although the terminus of SSG is at a higher elevation than AQK, and this phenomenon was concentrated from November to March (Figure 3a,b). The minimum and maximum diurnal temperature differences of AQK occurred on 2 June and 15 April, with differences of 2.9 °C and 26.5 °C, respectively. The minimum and maximum daily temperature differences of the terminus of SSG occurred on 2 June and 11 May, with differences of 3 °C and 15.8 °C, respectively (Figure 3c). In short, the diurnal temperature difference of the terminus of SSG has little fluctuation throughout the year, especially after entering the winter months. However, the diurnal temperature difference of AQK in the winter months is much higher than that in the summer months. During the summer months, the diurnal temperature difference in AQK is slightly greater than that at the terminus of SSG, but in the winter months, the diurnal temperature difference of the former is much greater than that of the latter.

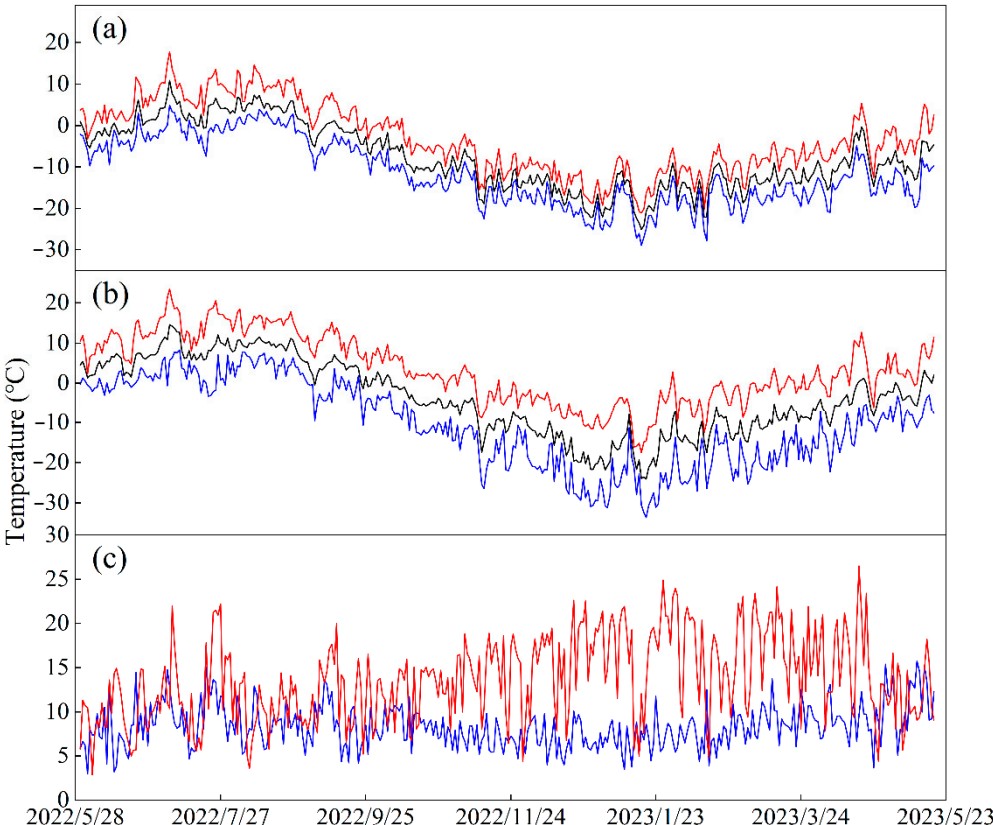

**Figure 3.** Temperature changes at the terminus of SSG (**a**) and AQK (**b**) (red line represents the daily maximum temperature; black line represents the daily average temperature; blue line represents the daily minimum temperature), and their maximum daily temperature differences (**c**) (red and blue lines represent the maximum daily temperature differences at AQK and the terminus of SSG, respectively).

The maximum monthly temperature differences of AQK and the terminus of SSG occurred in January and April, with differences of 36.5 °C and 26.3 °C, respectively; the minimum monthly temperature differences occurred in August, with differences of 17.4 °C and 16.6 °C, respectively. According to the temperature and altitude gradient of AQK and the terminus of SSG, we determined the vertical rate of temperature decrease. The results show that the Ulugh Muztagh Peak area has the characteristics of a small vertical rate of temperature decrease in the winter months (October–April) and a large vertical rate of temperature decrease in the summer months (May–September), which are −0.35 °C/100 m and −0.69 °C/100 m, respectively. However, there are significant differences in the temperature gradient change in different months, with the maximum value of the temperature gradient occurring in May at −0.86 °C/100 m, and the minimum value occurring in January at −0.02 °C/100 m, with a 42-fold difference between the two (Table 2).

**Table 2.** Monthly temperature difference of AQK and the terminus of SSG, and the vertical rate of temperature decline (VROTD) between them.

|  | January | February | March | April | May | June | July | August | September | October | November | December |
|---|---|---|---|---|---|---|---|---|---|---|---|---|
| AQK | 36.5 | 27.7 | 29.9 | 34.9 | 24.2 | 18.7 | 26.7 | 17.4 | 25.9 | 24.2 | 31.1 | 29.3 |
| SSG | 23.5 | 23.3 | 18.1 | 26.3 | 24.9 | 21.38 | 25.2 | 16.6 | 19.7 | 18.0 | 20.5 | 17.4 |
| VROTD | −0.02 | −0.32 | −0.43 | −0.57 | −0.86 | −0.68 | −0.66 | −0.59 | −0.64 | −0.63 | −0.43 | −0.09 |

The daily air temperature at the median height of SSG was calculated by using the vertical decline rate of air temperature at the two meteorological stations at AQK and the terminus of SSG. The daily air temperature values there took into account not only the vertical decline rate of air temperature, but also the cold storage of the glacier, and were

revised (Figure 4). The average daily temperature of AQK was positive from 15 May to 23 September. From 24 September to 6 October, the average daily temperature began to show positive and negative temperature fluctuations, and it entered a stable negative temperature period after 7 October (Figure 4a). Temperature monitoring at the meteorological station at the terminus of SSG showed that the daily average temperature in this area has a positive temperature period from 23 June to 2 September, accompanied by two fluctuation periods. From 15 June to 22 June is the fluctuation period before the establishment of the positive temperature period, and 3 September to 15 September is the fluctuation period before the establishment of the negative temperature period at the terminus of SSG (Figure 4b). At the median height of SSG, there were stable periods of positive temperatures from 23 to 24 June, 3 to 11 July, 23 to 31 July, and 4 to 27 August, with 42 days in the year when the daily mean temperature was greater than 0 °C, a positive cumulative temperature of 71 °C, an annual mean temperature of −11.4 °C, and an average temperature during the ablation period of 1 °C (Figure 4c).

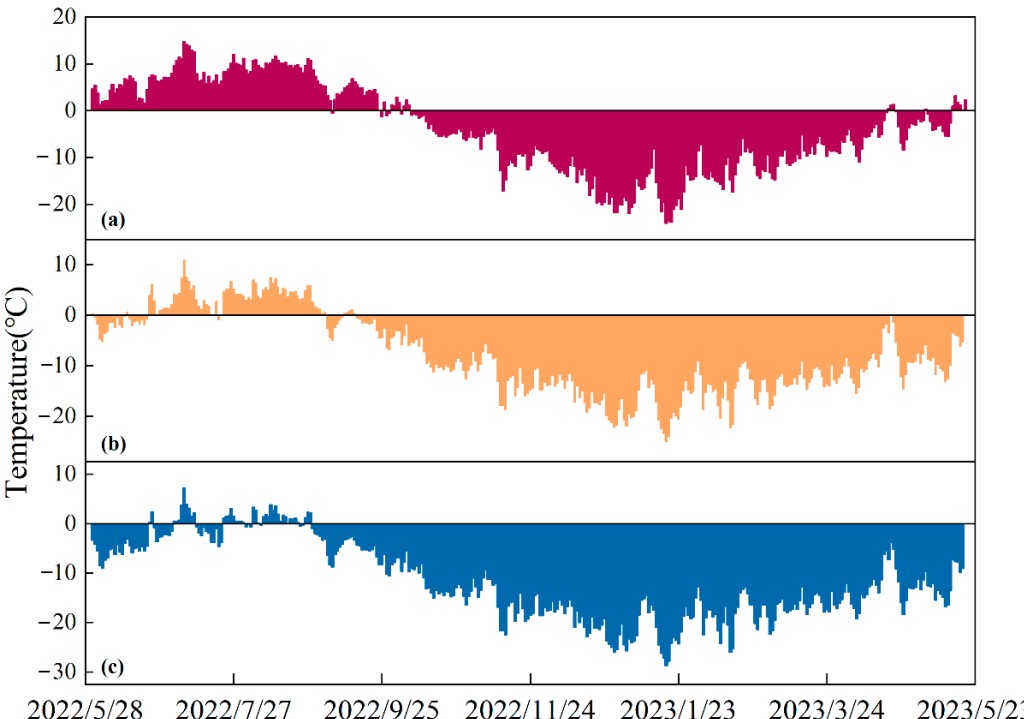

**Figure 4.** Diurnal temperature variations at AQK (**a**) and the terminus of SSG (**b**); the median height of SSG (**c**).

4.1.2. Solar Radiation

The diurnal variation trend of solar radiation at AQK and the terminus of SSG is essentially the same (Figure 5a,b). In the winter months, the amount of solar radiation is low and stable, while in the summer months, the amount of solar radiation is large but fluctuates greatly, and the average annual radiation amounts are 214 W/m$^2$ and 194 W/m$^2$, respectively. On the daily scale (Figure 5c), the daily maximum radiation at AQK and the terminus of SSG occurred at 13:00 p.m., with values of 732 W/m$^2$ and 663 W/m$^2$, respectively, and the solar radiation at AQK was greater than that at the terminus of SSG. The maximum transient radiation of AQK occurred at 13:00 on 25 June, at 1552 W/m$^2$, while the maximum transient radiation of the terminus of SSG occurred at 13:50 on 13 June, at 1593 W/m$^2$, which was higher than that of AQK. On an annual scale, the average monthly maximum radiation value of AQK occurred in July, at 290.9 W/m$^2$, the average monthly maximum radiation value of the terminus of SSG occurred in May, at 292.7 W/m$^2$, and the average monthly minimum radiation of the two places occurred in December, at

129.6 W/m² and 102.3 W/m², respectively. Except for May and June, the solar radiation of AQK was greater than that of the terminus of SSG in all other months, and the solar radiation was significantly affected by precipitation. In August, precipitation was greatest at the terminus of SSG. Cloudy and rainy weather results in solar radiation in August being at a radiatively low value relative to the summer as a whole (Figure 5d).

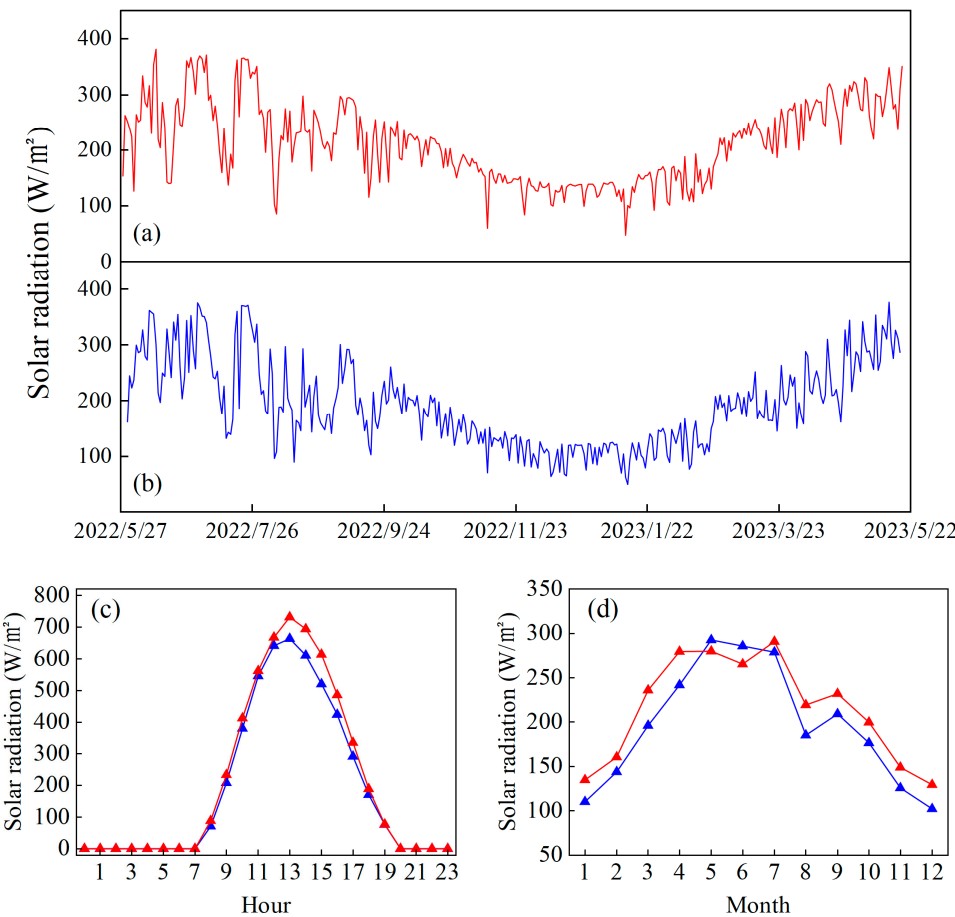

**Figure 5.** Changes in solar radiation at AQK (**a**) and the terminus of SSG (**b**); radiation changes at both sites on a daily scale (**c**) and on an annual scale (**d**), with the red and blue lines showing solar radiation changes at AQK and SSG, respectively.

### 4.1.3. Relative Humidity

The daily variation curve of relative humidity at AQK and SSG shows a single-peak and single-valley type of variation. The relative humidity is opposite to the daily variation in temperature, with the minimum occurring at midday, when solar radiation is stronger, the temperature is higher, the atmospheric saturated water vapor pressure is high, and the relative humidity is lower; the opposite is true in the early morning hours. In the afternoon, the daily change in relative humidity is a dry-to-wet process; the maximum daily relative humidity of AQK and the terminus of SSG appeared at 7:00 a.m. and 6:00 a.m., with values of 69.2% and 58.5%, respectively. The minimum relative humidity occurred at 16:00 and 15:00 in the late afternoon, with values of 40.6% and 46.6%, respectively (Figure 6a). On an annual scale (Figure 6b), the maximum relative humidity at AQK and the terminus of SSG both occurred in August, with values of 77.3% and 80.5%, respectively, and the minimum relative humidity occurred in March and January, with values of 45.8% and 39.9%, respectively. The changes in relative humidity were consistent with the changes in precipitation, with more precipitation and higher relative humidity in June and August, and less precipitation and significantly lower relative humidity in July, which is related to the temperature suppression during precipitation and the involvement of large amounts of

water vapor. The relative humidity of AQK in winter is higher than that at the terminus of SSG, which is mainly due to the higher wind speed of the glacier in winter and the increased air mobility at the terminus of the glacier. AQK is subject to the effects of evaporation and sublimation of the lake water and ice, and the relative humidity there in winter is relatively higher than that at the terminus of the glacier.

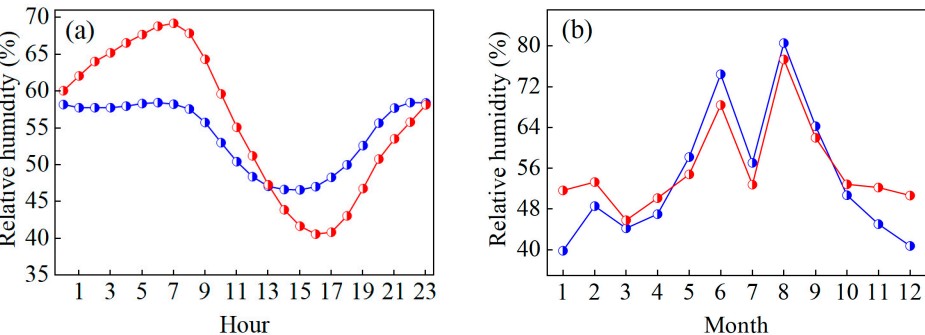

**Figure 6.** Daily-scale (**a**) and annual-scale (**b**) changes in relative humidity at AQK and the terminus of SSG; the red and blue lines are the relative humidity variations at AQK and SSG, respectively.

### 4.1.4. Precipitation and Rain–Heat Relations

In the mountainous regions of Western China, the trend of annual precipitation distribution is related to the proximity of water vapor transport, altitude, and topography [41]. Precipitation data recorded at the two weather stations (Figure 7a,b) showed that the terminus of SSG produced precipitation on a total of 114 days throughout the year, totaling 415.8 mm. The seasonal distribution of precipitation was summer (278.2 mm, 67%) > spring (62.4 mm, 15%) > fall (43.4 mm, 10%) > winter (31.8 mm, 8%), with the highest precipitation in the summer and the lowest in the winter, which is typical of summer-accumulation-type glaciers (Figure 7a). Due to the difficulty of recovering winter precipitation at the meteorological station at AQK, this study did not calculate the statistics for the whole year. The total number of days of precipitation from April to October at AQK was 61 days, with a cumulative precipitation of 183.4 mm (Figure 7b). In the east Kunlun Mountains area, precipitation was concentrated in May to September, both at AQK and at the terminus of SSG, with cumulative precipitation of 171 mm and 312.8 mm, respectively, and with the largest precipitation in August (77.2 mm and 177.8 mm, respectively), which accounted for 45.1% and 56.6% of the total precipitation from May to September, respectively. At AQK and the terminus of SSG, after the temperature reached its minimum in January, it began to rise slowly, and with the increase in temperature, precipitation began to increase in June–August, forming stable high-temperature weather from late July to early September. The precipitation also reached the extreme value, and the rainy heated period was significant. The maximum daily precipitation of AQK and the terminus of SSG was 16.6 mm and 22.8 mm, which occurred on 23 June and August 5, respectively. Precipitation patterns are influenced by air temperature. When the temperature is <2.5 °C, the precipitation is solid; when the temperature is >4 °C, it is liquid; and when the temperature is in between, it is mixed solid–liquid precipitation [42]. At the terminus of SSG, solid precipitation (227.8 mm), liquid precipitation (98.4 mm), and mixed solid–liquid precipitation (89.6 mm) accounted for 54.8%, 23.7%, and 21.5% of the total precipitation, respectively.

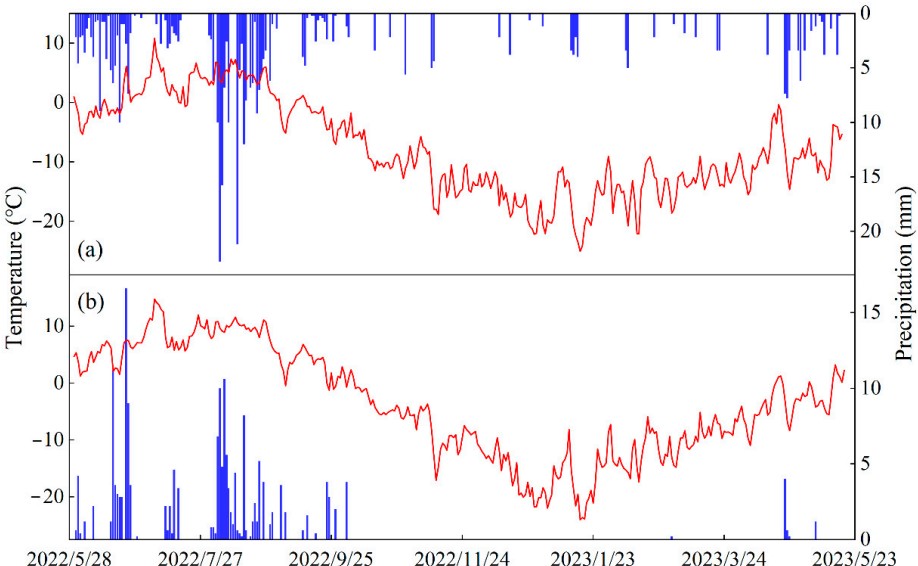

**Figure 7.** Changes in daily precipitation and air temperature at the terminus of SSG (**a**) and AQK (**b**); the red line is air temperature, and the blue bar is precipitation.

### 4.2. Momentum Conditions

#### 4.2.1. Wind Speed and Direction

AQK is consistent with the terminus of SSG on a daily scale, with significant daily variation in wind speed (Figure 8a). The wind speed at AQK started at 7:00 a.m. (0.9 m/s) and gradually intensified to reach the daily maximum at 18:00 p.m. (2.09 m/s); the wind speed at the terminus of SSG started at 9:00 a.m. (1.15 m/s) and gradually intensified to reach the daily maximum at 16:00 p.m. (2.15 m/s). Changes in wind speed are very similar to changes in air temperature, and it can be seen that, on small scales, differences in barometric pressure gradients caused by temperature changes drive wind speed changes. At the annual scale (Figure 8b), the maximum monthly average wind speed at AQK and the terminus of SSG occurred in April and January, with values of 1.81 m/s and 1.73 m/s, respectively. The monthly average minimum wind speeds of 1.08 m/s and 1.06 m/s occurred in December and August, respectively, and the annual average wind speeds were 1.44 m/s and 1.35 m/s, respectively. Combining the monthly average wind speeds with the hydrothermal conditions described above, we found the weather at AQK and the terminus of SSG to be breezy and hydrothermal, which did not seem to be prone to the formation of precipitation at higher wind speeds. The maximum instantaneous wind speed at AQK and the terminus of SSG occurred at 15:10 on 25 January and at 17:20 on 3 April, respectively, with values of 9.16 m/s and 8.02 m/s, respectively. The changes in wind speed in the two places show the change rule of alternating high and low values, with higher average wind speeds in January (the terminus of SSG), April, July, and November, and lower wind speeds in February-March, May–June, and August–October between these months. It is worth noting that there was a huge difference in wind speed between AQK and the terminus of SSG from December to March. Wind speeds at the terminus of SSG are much greater than those at AQK, and this is quite different from the wind speed variations in other months. In other months, the trend is essentially the same, and the difference in wind speed is not significant, although there is also a difference in wind speed between the two. Overall, the average daily and monthly wind speeds were not high at either site.

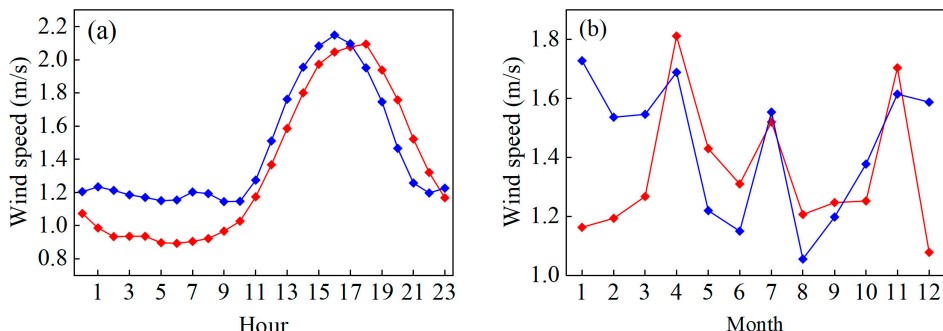

**Figure 8.** Wind speed variations at AQK and the terminus of SSG on the daily (**a**) and annual (**b**) scales; the red and blue lines are wind speed variations at AQK and the terminus of SSG, respectively.

### 4.2.2. Local Circulation

The development of local circulation at AQK and the terminus of SSG is significant (Figure 9 and Table 3). AQK has a significant lake effect and dominates the wind direction at the site. As can be seen from the diurnal winds at AQK in January and July, there was a clear diurnal shift in wind direction in both winter and summer. Westerly winds predominated during the day, with a frequency of 38.17% and 47.85% in January and July, respectively. The winds were predominantly easterly at night, with a frequency of 38.17% and 28.76% in January and July, respectively, and the wind speeds were greater in July than in January (Figure 9a,b). In addition to the easterly winds, the northeasterly winds also accounted for a large proportion of 25.27% during the night in July at AQK, which may have been caused by the shifting or unstable center of low pressure in the lake during the night.

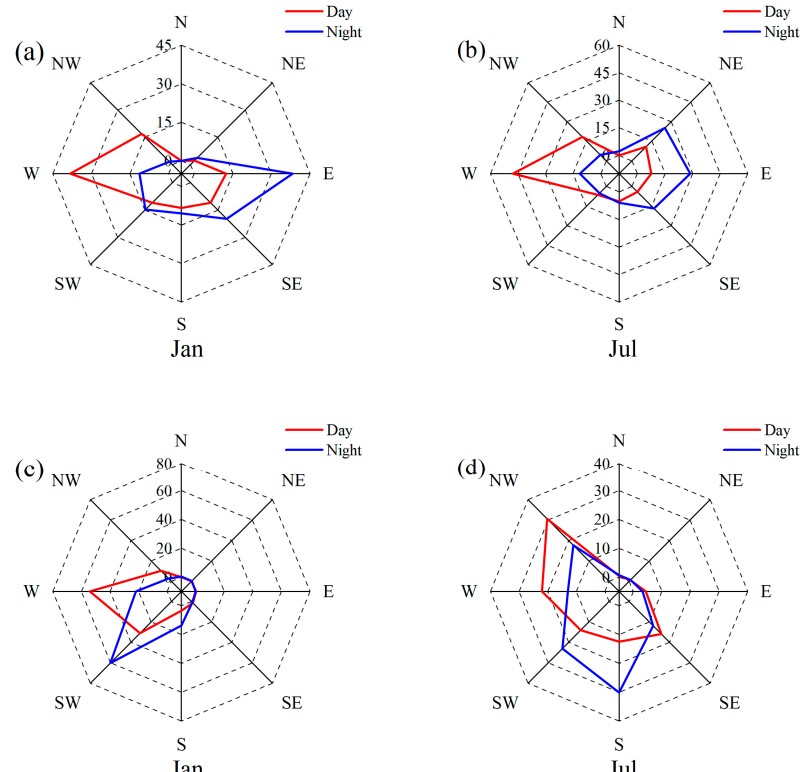

**Figure 9.** Changes in diurnal wind direction in January and July at AQK and the terminus of SSG: (**a**,**b**) are the wind frequencies of AQK in January and July, respectively; (**c**,**d**) are wind frequencies at the terminus of SSG in January and July, respectively.

**Table 3.** Wind direction and speed (WD and WS) in January and July for AQK and the terminus of SSG.

| Station | Time | WD/WS | N | NE | E | SE | S | SW | W | NW |
|---|---|---|---|---|---|---|---|---|---|---|
| AQK | Day of January | WD (%) | 0 | 2.15 | 12.37 | 11.02 | 8.33 | 11.02 | 38.17 ** | 16.94 |
| | | WS (m/s) | 0 | 1.84 | 0.81 | 0.52 | 0.58 | 2.02 | 1.77 | 1.43 |
| | Night of January | WD (%) | 0 | 3.76 | 38.17 ** | 19.89 | 10.48 | 14.78 | 11.29 | 1.61 |
| | | WS (m/s) | 0 | 0.89 | 0.77 | 0.64 | 0.66 | 1.54 | 1.49 | 1.21 |
| | Day of July | WD (%) | 0 | 10.75 | 7.53 | 4.03 | 5.11 | 6.45 | 47.85 ** | 18.28 |
| | | WS (m/s) | 0 | 2.23 | 1.26 | 1.04 | 1.07 | 1.05 | 2.04 | 1.58 |
| | Night of July | WD (%) | 2.15 | 25.27 | 28.76 ** | 16.94 | 5.91 | 5.11 | 11.29 | 4.57 |
| | | WS (m/s) | 2.03 | 1.5 | 0.97 | 0.78 | 0.92 | 1.05 | 2.1 | 2.11 |

| Station | Time | WD/WS | N | NE | E | SE | S | SW | W | NW |
|---|---|---|---|---|---|---|---|---|---|---|
| SSG | Day of January | WD (%) | 0 | 0 | 0.3 | 1.2 | 3.3 | 31 | 54 ** | 10.2 * |
| | | WS (m/s) | 0 | 0 | 1 | 0.4 | 1.1 | 1.9 | 2.5 | 1.4 |
| | Night of January | WD (%) | 0.3 | 0.3 | 0 | 0.7 | 13.6 * | 60.4 ** | 21.9 | 2.8 |
| | | WS (m/s) | 0.3 | 0.3 | 0 | 0.7 | 1 | 1.5 | 1.3 | 0.8 |
| | Day of July | WD (%) | 0.3 | 0.5 | 4.3 | 15.9 | 12.4 | 14 | 22* | 30.6 ** |
| | | WS (m/s) | 3 | 0.6 | 1.4 | 1.4 | 1.4 | 1.5 | 1.7 | 2.3 |
| | Night of July | WD (%) | 0.5 | 0.5 | 3.2 | 11.8 | 30.1 ** | 23.1 * | 12.9 | 17.7 |
| | | WS (m/s) | 0.8 | 1.2 | 1.2 | 1.7 | 1.2 | 1.3 | 1.1 | 1.6 |

Note: * main wind direction, ** dominant wind with greater frequency.

At the terminus of SSG on Ulugh Muztagh Peak, the river valley develops with a distinct valley form. In January, the prevailing westerly winds dominated the SSG both during the day and at night. Westerly wind speeds were greater during the day (2.2 m/s) than at night (1.4 m/s), and valley winds influenced by the Bureau Circulation were not significant, with northwesterly winds during the day (10.2%) and southerly winds at night (13.6%) (Figure 9c). In July, the development of valley winds was significant. The course of SSG means that the mountain winds blowing down to the valley are predominantly southerly, while the upward valley winds are predominantly northerly. During the summer months, a westerly wind system still affects the Ulugh Muztagh region, with westerly winds (22%) during the day and southwesterly winds (23.1%) during the night, but they do not predominate, with the dominance of diurnal wind shifts influenced by local circulation: northwesterly winds during the day (30.6%), and southerly winds at night (30.1%). On the other hand, SSG also had more pronounced glacial winds (12.4%) during the hot daytime hours of July, influenced by the underlying surface of the glacier (Figure 9d).

*4.3. Mass Balance and Simulation of Glacier Ablation Depth in SSG*

4.3.1. Mass Balance in the Ablation Zone of SSG

During the 30 May 2022 expedition, range poles could not be placed higher up SSG due to the harsh natural conditions in the glacier area; all six range poles were placed in the ablation zone of the glacier. On 18 May 2023, the snow was 25–30 cm thick on the ice at the terminus of SSG, and during the range pole data measurements, pole A, with the lowest elevation, was not measured because it fell; the results of the remaining five range poles are shown in Table 4.

**Table 4.** Mass balance in the ablation zone of SSG.

| Name | Latitude | Longitude | Altitude (m) | Mass Balance (mm) |
|---|---|---|---|---|
| B | 36.599 | 87.381 | 5200 | −1747 |
| C | 36.597 | 87.382 | 5220 | −1816 |
| D | 36.595 | 87.383 | 5222 | −1616 |
| E | 36.592 | 87.384 | 5234 | −1796 |
| F | 36.591 | 87.385 | 5302 | −1956 |

The average glacial mass balance in the ablation zone of SSG was −1766 mm, and the change in the mass balance value with altitude was not regular, which may have been related to the closer proximity of the range poles and differences in topographic shading. Although it was not possible to measure the mass balance value near the equilibrium line due to environmental factors, the measured mass balance values of the glacier in the ablation zone of SSG are still rare and valuable data. This set of values fills a gap in observations of glacial mass balance in this region.

4.3.2. Deep Simulation of Glacial Ablation

Glacier ablation and accumulation is an indispensable part of the study of glacier mass balance, which is of great significance to the understanding of glaciers' development, advance, and retreat changes. We calculated the vertical rate of decrease in air temperature at AQK and the terminus of SSG using air temperature data from the two stations, from which the average air temperature from June to August at the height of the median area of SSG was calculated to be −1.13 °C. The median area height of SSG is 5531.2 m a.s.l according to the Second Glacier Inventory data from China. Through the statistics of hourly and daily air temperature data at the terminus of SSG, the ablation period of SSG was circled from mid-June to mid-September, the average daily air temperature at the median height of the ablation period was 1 °C, and the radiation balance value was taken as 0.93. $DDF_{ice}$ was 8.7 mm/d/°C and $DDF_{snow}$ was 8.26 mm/d/°C at the median altitude of SSG, as calculated by the degree-day factors conversion formula. Following the principle that the height of the median glacier area is the criterion for determining the height of the mass balance line, and that ablation is equal to accumulation, only the snow degree-day factor was used in the calculation process. The positive cumulative temperature statistic at the median altitude was 71 °C. Based on the above data and methods, the average annual ablation depth of SSG is shown in Table 5.

**Table 5.** Simulation results of glacial ablation of SSG.

| Methods | References | Results |
|:---:|:---:|:---:|
| Climate coefficient method | [38] | 597 mm/a |
| Glacier zero-balance line method | [39] | 596 mm/a |
| Degree-day factors method | [40] | 586 mm/a |

Based on the calculations, the region has a larger value of degree-day factors. The reason for this is that at the higher altitude of the glacier area, solar radiation is more intense, and the main source of energy for glacier ablation is solar radiation, which leads to an increase in glacier ablation. Higher-altitude glacial areas have lower mean daily temperatures, resulting in smaller positive cumulative temperatures, so this region has a higher value of degree-day factors. Based on the above data and methods, the average ablation depth of SSG was derived to be approximately 587–597 mm/a.

## 5. Discussion

### 5.1. Relationship between Temperature and Terrain

In the east Kunlun Mountains, we collected the temperature data for AQK and the terminus of SSG, and we found that the daily temperature difference at the terminus of SSG fluctuates little throughout the year, especially in the winter half of the year, when the daily temperature difference is more stable. However, the diurnal range of AQK is the largest in the winter half of the year (Figure 3b), and the minimum temperature is lower than that at the higher elevation of the terminus of SSG, which is the most obvious in the winter half of the year (Figure 10). This is related to the topography of the two places. AQK is located in a basin, with high elevations on all sides and low elevations in the middle. At night, the heat dissipation rate of the surrounding highlands is fast, forming cold air, while the heat dissipation in the middle is slow, forming warm air. The

air pressure in the surrounding highlands is higher than that in the central lowlands, and cold air from the highlands flows and accumulates towards the lowlands. As the warm air in the central lowlands rises, it generally does not form clouds, because the altitude is not high, so the surface air temperature of the lowlands decreases. The higher the elevation of the surrounding highlands, the lower the temperature of the flowing cold air, which makes the temperature of the lowlands lower than that of the surrounding highlands from night until dawn. During the day, because the highlands are heated quickly and the lowlands are heated slowly, the cooler air in the lowlands flows to the surrounding highlands, and the warmer air over the lowlands sinks and replenishes. As a result, the temperature in the lowlands is higher than that in the surrounding highlands during the day, so the diurnal range in the lowlands is greater than that in the highlands. This phenomenon is more pronounced in winter, when temperatures are lower in the surrounding highlands.

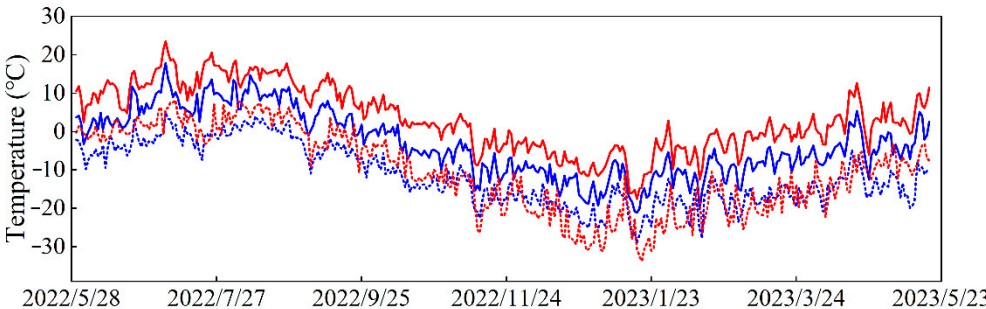

**Figure 10.** Maximum and minimum temperatures of the terminus of SSG and AQK. The red solid line and dotted line are the daily maximum and minimum temperature of AQK, respectively. The blue solid line and dotted line are the daily maximum temperature and daily minimum temperature at the terminus of SSG, respectively.

### 5.2. The Relationship between Meteorological Elements in the East Kunlun Mountains

Above, we focused on the analysis of the time-series changes of the meteorological elements monitored by the two meteorological stations in east Kunlun. Through the comparison and analysis of the results, we can see that there is a strong correlation between some meteorological elements. Here, we further discuss and analyze the relationship between the various meteorological elements at AQK and the terminus of SSG. On the daily scale (Figure 11a,b), the relative humidity and temperature, solar radiation and relative humidity, and wind speed and relative humidity at AQK and at the terminus of SSG were negatively correlated, passing the significance test of 0.01. This is because, on the daily scale, high temperatures lead to high saturated water vapor pressure and low relative humidity, and they are negatively correlated. On the other hand, the change in wind speed driven by temperature accelerates the flow of air, which is not conducive to the collection of water vapor. The greater the wind speed, the lower the relative humidity on the daily scale. In addition to this, the relationship between wind speed and relative humidity is indirectly affected by the relationship between temperature and relative humidity. Solar radiation and temperature, wind speed and temperature, and wind speed and solar radiation are all positively correlated, and all of these pairs passed the significance test, indicating that on the daily scale, the pressure gradient generated by the temperature change drives the change in wind speed, and this law is extremely significant for the daily change (Figure 8a). The correlation between precipitation and other meteorological elements is not obvious, because the precipitation in both places is concentrated in summer, and the variation law of precipitation in the year is diluted statistically on the daily scale.

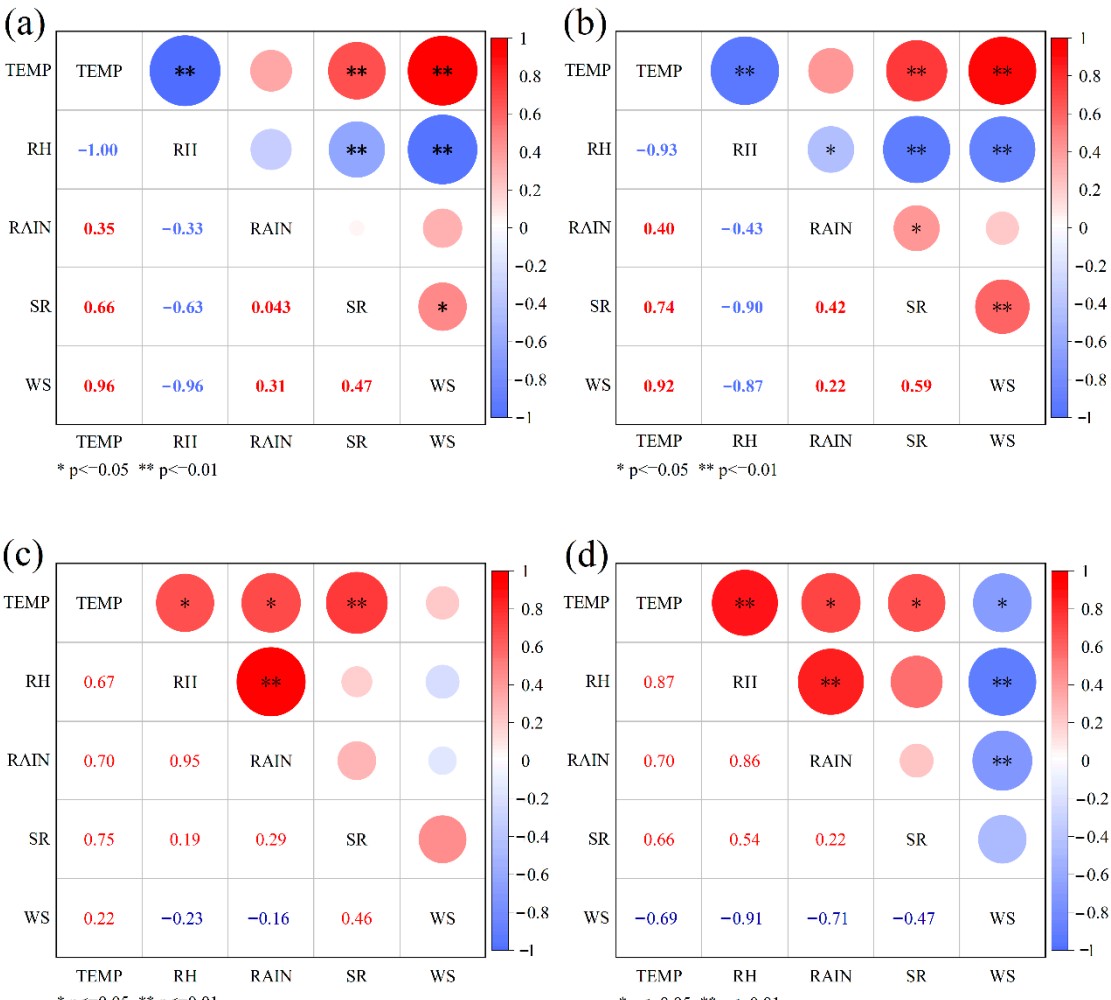

**Figure 11.** The relationship between meteorological elements at AQK and the terminus of SSG: (**a,b**) the correlation between meteorological elements at the diurnal scale at AQK and the terminus of SSG, respectively. (**c,d**) The correlation between meteorological elements at the annual scale at AQK and the terminus of SSG, respectively.

Through statistical analysis on the annual scale (Figure 11c,d), it can be found that there are positive correlations between air temperature and relative humidity, precipitation and air temperature, precipitation and relative humidity, and solar radiation and air temperature at AQK and the terminus of SSG. In meteorology, a single temperature has an inverse relationship with relative humidity, but in the east Kunlun Mountains from June to September, convective precipitation caused by high-temperature weather increases the relative humidity of the air, so the relative humidity and temperature are positively correlated, and the two are only indirectly positively correlated based on precipitation. This is different from the statistics on the daily scale, because the variation law of precipitation is downplayed in the daily-scale statistics, the singleness of temperature and relative humidity is more significant, and they are negatively correlated. At the annual scale, there is no significant correlation between wind speed and the meteorological elements at AQK (Figure 11c), but at the terminus of SSG, wind speed is negatively correlated with temperature, relative humidity, and precipitation, and it passes the significance test (Figure 11d). It can be seen that the influence of wind speed at the terminus of SSG on other meteorological elements is greater than that at AQK. The reason for this is that the high wind speed at the terminus of the glacier accelerates the flow of air, which is not conducive to the collection of water vapor, resulting in less precipitation and low relative humidity. The negative correlation between wind speed and temperature is mainly influenced by circulation on a larger scale.

In winter, the area is controlled by the westerly wind circulation, with high wind speeds. However, in summer, when the temperature is higher, the wind speed is lower due to the influence of the depression system and local circulation in the plateau. Therefore, the negative correlation between wind speed and temperature reflects the relationship on a larger spatial and temporal scale, which is contrary to the statistical results on the daily scale. There is no significant correlation between wind speed and other meteorological elements in the AQK region (Figure 11c). This suggests that relatively enclosed terrain is more affected by wind speed, and that fine-scale statistics are better at portraying the relationships between meteorological elements.

In order to further explore the relationship between daily temperature difference and precipitation, we compared the data of temperature and precipitation and found that the reason for the small daily temperature difference was mainly the small amount of solar radiation, caused by rainy weather and the low daily maximum temperature, which ultimately led to the small daily temperature difference, while the weather conditions during the period of large temperature difference were the opposite (Table 6). As can be seen from Table 6, the reason for the smallest daily temperature difference was that precipitation inhibited the hot weather after noon. Through comparison of precipitation data, we found that although the daily precipitation of 2.2 mm or 0.4 mm was not the maximum daily precipitation, it effectively inhibited the maximum temperature of the day, because precipitation occurred within a few hours after the formation of the afternoon peak temperature. The precipitation at AQK and the terminus of SSG on June 2 occurred from 17:00 to 18:00 and from 14:00 to 16:00 Beijing time, respectively. In rainy weather, thick clouds are extremely effective in blocking solar radiation, resulting in less obvious ground warming and smaller daily temperature differences. In the sunny and cloudless afternoon, due to the absence of cloud cover, the solar radiation value is large; the ground temperature rises quickly and is high in the daytime, while it drops quickly and is low at night, and the daily temperature difference is large. This is different from Liu's [43] results on the influence of summer clouds on the ground in the east of Rongbuk Glacier on the north slope of Mount Everest. In Liu's study, increased long-wave radiation from monsoon clouds accelerated the glacier ablation, while in this study, the cloud cover in the SSG area reduced the total solar radiation received by the ground, suppressed the high-temperature weather and, thus, reduced the melting of glaciers.

**Table 6.** Mean daily temperatures and temperature differences of AQK and the terminus of SSG.

| Name | Element | Temperature (°C) | Time (Month/Day/Hour) | Weather | Precipitation (mm) | Radiation (W/m²) |
|---|---|---|---|---|---|---|
| AQK | Daily maximum temperature | 23.49 | 7/6/16 | Sunny | 0 | 380 |
| | Average daily maximum temperature | 14.6 | 7/6 | Sunny | 0 | 570 |
| | Average daily minimum temperature | −24 | 1/19 | Sunny | 0 | 298 |
| | Daily minimum temperature | −33.62 | 1/19/7 | Sunny | 0 | 0 |
| | Maximum daily temperature difference | 26.53 | 4/15 | Sunny | 0 | 636 |
| | Minimum daily temperature difference | 2.86 | 6/2 | Rainy | 0.4 | 225 |
| SSG | Daily maximum temperature | 17.81 | 7/6/15 | Sunny | 0 | 763 |
| | Average daily maximum temperature | 10.9 | 7/6 | Sunny | 0 | 607 |
| | Average daily minimum temperature | −25 | 1/17 | Sunny | 0 | 204 |
| | Daily minimum temperature | −28.92 | 1/17/9 | Sunny | 0 | 9.36 |
| | Maximum daily temperature difference | 15.76 | 5/11 | Sunny | 0 | 617 |
| | Minimum daily temperature difference | 2.96 | 6/2 | Rainy | 2.2 | 224 |

*5.3. Comparison of the Ablation Depths of SSG with Other Glaciated Areas on the TP*

The depth of glacial ablation depends on factors such as the geographic location of the glacier, its elevation, climatic conditions, and the type of glacier. Marine-type glaciers are subject to low latitudes, high temperatures, and abundant precipitation, with strong glacial ablation and maximum multiyear average glacier ablation depths. Conversely, continental-type glaciers, with higher latitudes, low temperatures, and sparse precipitation, have relatively weak glacial ablation and small average ablation depths. In this study, we

compared the average annual ablation depths of other glacial action areas, and as shown in Table 7, as latitude increases, the climate becomes more continental, and the glacier ablation depth decreases significantly. The regional distribution of glacier ablation depth is highly correlated with the regional distribution of precipitation and temperature. However, the nature of the ice surface is also one of the main factors affecting glacier ablation, and the thickness of localized glacier surface moraine cover can change the consistency of the glacier and the climate distribution [44–46]. Compared with other glacial areas in Western China, SSG and the western end of the Qilian Mountains are closer in climate zoning; the simulated ablation depth of SSG is also closer to that of the western Qilian Mountains, and both have the characteristics of high mass balance line altitude, low precipitation, and large amounts of radiation, which are typical features of continental glaciers. Compared with the ablation depth of other glaciers, it can be seen that the simulated ablation depth of SSG is consistent with the change law of glacier ablation depth by climate zone. This shows that the results of glacier ablation simulated by the empirical formula used in this study are well referenced. It should be noted that the depth of glacier ablation changes in response to climatic fluctuations, the magnitude of which depends on the level of temperature and the amount of precipitation. Summer temperatures and precipitation conditions directly affect snowline heights, leading to changes in the intensity of glacier ablation.

**Table 7.** Average annual ablation depth of SSG and other glaciers in Western China.

| Glaciers in Western China | Average Annual Ablation Depth (mm) | References |
| --- | --- | --- |
| Guxiang Glacier in Bomi, Tibet | 2678 | [47] |
| Rongbuk Glacier in the Central Himalaya | 660 | [48] |
| Northern slopes of the western Tian Shan | 1600 | [48] |
| Northern slopes of the eastern Tian Shan | 600 | [47] |
| Southern slopes of the eastern Tian Shan | 1000 | [49] |
| Western part of the Qilian Mountains | 650 | [48] |
| Middle part of the Qilian Mountains | 900 | [48] |
| Eastern part of the Qilian Mountains | 1200 | [48] |
| Western Kunlun (south of the Yulong Kashi River) | 200 | [50] |
| North slope of Ulugh Muztagh Peak, east Kunlun * | 587–597 | This study |

Note: * the study area of this study.

### 5.4. Limitations and Implications

In this study, we used about 1 year of measured meteorological data from meteorological stations erected on the east shore of AQK and the terminus of SSG, along with range pole data from the terminus of the SSG ablation zone from May 2022 to May 2023. We conducted a statistical analysis of the micrometeorology of the two areas and the mass balance of the terminus of the glacier ablation zone, as well as a simulation of the glacier ablation. The shortcoming of this study is that the time scale of the meteorological data of the two weather stations is relatively short, with only 1 year of data, and the results and analyses may have peculiarities that deviate from the mean state. Especially in terms of wind speed, the changes manifested in the time series are difficult to interpret, requiring continuous observation at a later stage, and the change law may become more stable when the amount of data accumulates to a sufficient level. Another point is that the precipitation data in winter are limited by the function of the instrument, which is not resistant to cold and freezing; additionally, the statistics are not accurate enough, and there may be a large error (we set up an extra T200B precipitation monitoring instrument at the terminus of SSG during our second field trip in May 2023 to minimize errors from winter precipitation). However, in May–September, when temperatures are high, the instrument's functionality is not affected, and the precipitation data are credible and precise. Precipitation from May to September reached 183.4 mm and 312.8 mm at AQK and the terminus of SSG, respectively, proving that precipitation in mountainous areas is seriously underestimated by model simulation data [51]. This has positive implications for the assessment and revision of

reanalysis data. The accuracy of reanalysis precipitation data is affected by season and altitude, and the higher the altitude, the worse the accuracy. For example, the evaluation of ERA5 data in the Chinese region shows that the precision of precipitation in the western region is less than that in the eastern region, and the precision of precipitation in the high-elevation region is lower than that in the low-elevation region. Compared with the winter and spring seasons, the accuracy for the summer and fall seasons is low [52]. The reason for this is that the accuracy of reanalysis data is affected by the quality of the original input data information and the assimilation algorithm, and the data quality of the reanalysis products is also relatively low in places with sparse observational data or in regions with low-quality input data [53]. In the simulation of glacier ablation, the relationship between air temperature and glacier ablation is utilized for simulation with empirical formulae, bringing the advantage of the simplicity and availability of the formulae, but also causing unavoidable errors. In the simulation of glacier ablation, due to the insufficiency of observation data, the effects of wind-blown snow, liquid precipitation, and refreezing processes on the ablation of snow and ice are not considered. These refined simulations are very much needed when the instrumentation and data are sufficient. Unfortunately, the harsh environment of the glacier region makes it difficult to place ranging poles at greater distances and over greater elevation gradients for mass balance monitoring. However, the first glacier mass balance monitoring using the range poles method in the east Kunlun maximum glacier action area is of very positive significance, and the mass balance data in the ablation area also provide us with an approximate picture of the glacier ablation in the region. Overall, this study reveals the finer micrometeorological characteristics of the east Kunlun Mountains, as well as the fact that precipitation in the mountains is underestimated. In the area of glacier ablation monitoring, the mass balance data of the ablation zone of SSG were obtained, and the average annual ablation depth of SSG was simulated. This will provide a good reference and experience for subsequent research on the east Kunlun Glacier.

## 6. Conclusions

Using measured meteorological data from 30 May 2022 to 18 May 2023 from meteorological stations on the east shore of AQK and at the terminus of SSG, along with mass balance data in the ablation area at the terminus of SSG, we analyzed the micrometeorological characteristics of the east Kunlun Mountains and conducted ablation simulations of SSG. The following realizations were gained:

(1) The temperatures at AQK and SSG have significant daily and seasonal variations, and the timing of precipitation has a large impact on the daily temperature difference. The average annual temperatures of the two places are $-3.7\,^{\circ}$C and $-7.7\,^{\circ}$C, respectively, with the highest monthly temperature occurring in August and the lowest in January, and the vertical rate of temperature decrease is characterized by a large summer half-year and a small winter half-year.

(2) The solar radiation of the two places is greater in the summer than in the winter, which is significantly affected by precipitation. The relative humidity is greater in the early morning than in the late afternoon, greater in the summer than in the winter, and shows a single-peak, single-valley type of change, which is mainly affected by temperature and precipitation.

(3) The maximum wind speeds at AQK and the terminus of SSG can reach 9.16 m/s and 8.12 m/s, respectively, but the monthly and annual average wind speeds are not high. Wind speed has a greater effect on other meteorological factors at the terminus of SSG than at AQK. On a daily scale, wind speed is driven by air temperature.

(4) By counting the frequency of diurnal winds, it was found that local circulation developed at both sites. The area around AQK is subject to lake effects throughout the year, while at the terminus of SSG, valley winds develop only in the summer and are controlled by the westerly belt in the winter.

(5) The cumulative precipitation from April to September in AQK was 183.4 mm, and the annual precipitation at the terminus of SSG glacier was 415.8 mm, which was mainly

concentrated in the summer, and the precipitation and heat were significant at the same time.

(6) In terms of glacier ablation monitoring, the average mass balance value of the ablation area of SSG, as measured by the range poles method, was −1786 mm, and the average annual ablation depth simulated by the empirical formula was 587–597 mm, which is not large compared with other glacier areas of the Tibet Plateau and has the characteristics of typical continental-type glaciers.

This study provides new insights into micrometeorological changes in the east Kunlun Mountains and the depth of glacial ablation in SSG. In order to further improve the monitoring of meteorological elements and the glacier mass balance monitoring network in the east Kunlun Mountains, it is necessary to build more meteorological monitoring stations, collect meteorological data over longer time scales, and monitor more range poles towards the higher altitudes of the glacier. This would provide more stable and refined data support for mountain meteorological observations and glacier mass balance observations.

**Author Contributions:** Methodology, M.S.; software, W.W.; formal analysis, S.N. and W.W.; investigation, Y.C.; data curation, W.W.; writing—original draft preparation, W.W.; writing—review and editing, X.Y.; project administration, M.Z. and Y.C.; funding acquisition, M.S. All authors have read and agreed to the published version of the manuscript.

**Funding:** This research was funded by the Third Xinjiang Scientific Expedition Program (No.2021xjkk0101) and the National Natural Science Foundation of China (No.42161027).

**Data Availability Statement:** The data that support the findings of this study are available from the corresponding author upon reasonable request.

**Conflicts of Interest:** The authors declare no conflict of interest.

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
