# Peer review of "Micrometeorological Analysis and Glacier Ablation Simulation in East Kunlun"

_water, doi:10.3390/w15193517_

Round 1
Reviewer 1 Report
This study was based on high-altitude meteorological stations placed on the eastern shore of Aqikkule Lake and at the terminus of Shenshechuan Glacier and on mass balance data from that glacier. The characteristics and correlation of temperature, solar radiation, relative
humidity, precipitation, wind speed and direction of the two regions and the
mass balance in the ablation area of Shenshechuan Glacier from May 30, 2022 to May 18, 2023 were analyzed. In addition, the average melting depth of Shenshechuan Glacier was simulated.
- The manuscript is clearly presented and presented in a well-structured manner.
- The cited references are relevant.
- The manuscript scientifically sound and the experimental design is appropriate.
- The results presented in the manuscript are reproducible based on the details given in the methods section
- The figures and tables are appropriate and properly show the data. The data are interpreted appropriately and consistently throughout the manuscript.
- The conclusions are consistent with the evidence and arguments presented.
- Specific suggested changes in the manuscript: Line 66: Please state the objectives of this study here. Line 123: The figure legend must be on the same page as the figure. Line 298: Relative humidity Line 385: Local circulation Line 594: Limitations and implications
The English must be checked and corrected by a native English person who knows the subject of research in this manuscript.
Author Response
Dear reviewer,
Thank you for recognizing and suggesting this study, your suggestions and corrections helped a lot to enhance and improve this article. I have revised the article according to your requests and suggestions.

Reviewer 2 Report
The paper includes conclusions based on information from meteorological stations and mass balance data and covers micrometeorological analysis and glacier ablation simulation in East Kunlun.
The paper's key conclusions are as follows:
1. In the research region, it was discovered that temperature and solar radiation were the primary influences on glacier ablation. Due to their accessibility and simplicity, the authors used empirical methods to model glacier ablation in the research region. I advise the writers to include a summary of the other approaches.
2. Lower altitude locations saw a higher rate of glacier ablation than higher altitude regions. Could you elaborate on what the literature claimed?
3. The authors suggest that to improve the monitoring of meteorological elements and the glacier mass balance monitoring network in the East Kunlun Mountains, it is necessary to build more meteorological monitoring stations, collect meteorological data on longer time scales, and monitor more range poles towards the higher altitudes of the glacier. This will provide more stable and refined data support for mountain meteorological observations and glacier mass balance observations. Could the authors give a proposed meteorological monitoring stations network based on the lack of information?
Author Response

(The authors gave the same response as above.)

Reviewer 3 Report
Conducting meteorological monitoring in high-altitude mountainous areas is very costly in terms of human and financial resources as well as high safety risks. This is one of the reasons why mountain weather research relies heavily on reanalysis of data. I am very supportive of this type of monitoring activity and the writing of articles for publication. However, the following points need to be considered and corrected in this article:
1. line 95 line 99 line 113 Note the standardization of the writing of units
2. lines 24 and 95-96, it is mentioned that the East Kunlun is under the control of the westerly wind belt, but some studies on the plateau monsoon have shown that the prevailing winds inside the plateau are greatly influenced by the seasonal high and low pressure variations in the plateau itself, and these findings and opinions should be taken into consideration.
3. Line 281, it is written that the maximum instantaneous radiation of AQK and SSC is 1552w/m2 and 1593W/m2 respectively, which is greater than the solar constant of 1367w/m2, that is not reasonable, please check your data.
4. The wind speed in the mountainous areas in the thesis is generally low, which is an important feature, but you need to consider whether the weather station data records have been interfered with, and the intuition at the place of study can be used as a reference.
5. The precipitation in the East Kunlun Mountains recorded by meteorological stations is much larger than some reanalysis data in the region, these meteorological monitoring data is a very important evidence, the paper lacks and other reanalysis of precipitation data to highlight these differences, it is recommended to cite a number of East Kunlun reanalysis of precipitation data in the paper, or a direct comparison of the data.
On the basis that the above suggestions are corrected and amended, I agree with the publication of this paper.
Author Response

(The authors gave the same response as above.)
